# Swelling, Protein Adsorption, and Biocompatibility In Vitro of Gel Beads Prepared from Pectin of Hogweed *Heracleum sosnówskyi Manden* in Comparison with Gel Beads from Apple Pectin

**DOI:** 10.3390/ijms23063388

**Published:** 2022-03-21

**Authors:** Sergey Popov, Nikita Paderin, Daria Khramova, Elizaveta Kvashninova, Olga Patova, Fedor Vityazev

**Affiliations:** Institute of Physiology of Federal Research Centre “Komi Science Centre of the Urals Branch of the Russian Academy of Sciences”, 50, Pervomaiskaya Str., 167982 Syktyvkar, Russia; paderin_nm@mail.ru (N.P.); dkhramova@gmail.com (D.K.); kvashninova.e@yandex.ru (E.K.); patova_olga@mail.ru (O.P.); rodefex@mail.ru (F.V.)

**Keywords:** hogweed pectin, apple pectin, gel beads, swelling, protein adsorption, complement activation, hemolysis, peritoneal macrophages, TNF-α

## Abstract

The study aims to develop gel beads with improved functional properties and biocompatibility from hogweed (HS) pectin. HS4 and AP4 gel beads were prepared from the HS pectin and apple pectin (AP) using gelling with calcium ions. HS4 and AP4 gel beads swelled in PBS in dependence on pH. The swelling degree of HS4 and AP4 gel beads was 191 and 136%, respectively, in PBS at pH 7.4. The hardness of HS4 and AP4 gel beads reduced 8.2 and 60 times, respectively, compared with the initial value after 24 h incubation. Both pectin gel beads swelled less in Hanks’ solution than in PBS and swelled less in Hanks’ solution containing peritoneal macrophages than in cell-free Hanks’ solution. Serum protein adsorption by HS4 and AP4 gel beads was 118 ± 44 and 196 ± 68 μg/cm^2^ after 24 h of incubation. Both pectin gel beads demonstrated low rates of hemolysis and complement activation. However, HS4 gel beads inhibited the LPS-stimulated secretion of TNF-α and the expression of TLR4 and NF-κB by macrophages, whereas AP4 gel beads stimulated the inflammatory response of macrophages. HS4 gel beads adsorbed 1.3 times more LPS and adhered to 1.6 times more macrophages than AP4 gel beads. Thus, HS pectin gel has advantages over AP gel concerning swelling behavior, protein adsorption, and biocompatibility.

## 1. Introduction

Polysaccharides isolated from plants are of great interest in the development of gel biomaterials with high biocompatibility [1,2]. Pectin is an anionic polysaccharide found in the cell wall and intracellular space of most flowering plants. Gelling capacity and low production costs ensure the wide use of pectin in the food and pharmaceutical industries [3,4]. Gel materials based on commercial pectins have been proposed for tissue engineering [5], wound dressing [6,7], drug delivery [8], and other biomedical applications [4]. However, a high degree of swelling, weakening of mechanical properties, and rapid degradation limit the usage of pectin gels as biomaterials. To overcome the mentioned drawbacks, chemical functionalization or blending of pectin with various polymers to improve its properties may be necessary [3,9,10].

The functional properties of the pectin hydrogel are determined by the structural features of the pectin macromolecule. Therefore, the mentioned limitations may be due to the chemical structure of commercial pectins. Traditionally, pectin hydrogels are derived from pectins extracted from citrus peel and apple pomace, which consist primarily of a linear chain of 1,4-α-D-galacturonic acid (GalA) residues. However, many natural pectins have a much more complex structure. Generally, pectic polysaccharides consist of structural domains of homogalacturonan (HG) containing GalA with some of the carboxyl groups as methyl esterified and acetylated forms; rhamnogalacturonan-I (RG I) having a backbone of repeating disaccharide of GalA and rhamnose with the side chains of arabinan, galactan, and arabinogalactan; rhamnogalacturonan-II (RG II) consisting of a backbone of HG with complicated side chains containing simple sugars linked to the GalA; xylogalacturonan (XG) [11]. Depending on the degree of methyl esterification (DM), pectins are divided into high-methyl esterified pectins (DM > 50%), which can form physical gels at pH < 3.5 n the presence of co-solutes, and low-methyl esterified pectins (DM < 50%) that can form gels in the presence of divalent cations in a wide pH range [12]. The properties of pectin gel crosslinked with Ca^2+^ ions can be controlled by pH, other monovalent counterions, and temperature. Therefore, pectin is suitable for externally tunable systems.

In general, pectin-containing scaffolds are biocompatible [13,14,15,16,17,18]. However, non-specific adsorption of blood proteins [19], including proteins of the complement system [20,21,22], and stimulation of macrophages to secrete pro-inflammatory cytokines and nitric oxide [23,24,25,26] has been reported with pectin materials. Non-specific protein adsorption, complement activation, and inflammation develop a foreign body response (FBR), which hinders the development of tissue engineering scaffolds due to the failure of integration of the implant with native tissue [27]. Therefore, the targeted design of new biomaterials based on pectin appears to require pectin characteristics that provide controlled swelling and degradation in physiological media and reduction of FBR.

We previously isolated RG-I containing pectin from hogweed *H. sosnówskyi* (HS), which had a low DM and formed strong gels in the presence of calcium ions [28]. Differences in rheological and textural properties were observed in gels prepared from hogweed pectins of different chemical compositions and molecular weights (Mw). HS pectin gel may be considered a promising biomaterial for tissue engineering because the HS gels were stronger than the gel obtained from the commercial apple pectin [28]. However, the degradation and swelling of the HS hydrogel in physiological solutions and its effect on FBR have not been investigated yet.

The study aims to develop gel beads with improved functional properties and biocompatibility from HS pectin. For this, hydrogel beads were prepared by simple dropwise adding solutions of commercial apple pectin (AP) and HS pectin to calcium chloride solution. The advantages of HS pectin gel compared to AP gel are demonstrated concerning swelling behavior, protein adsorption, and inflammatory macrophage activation.

## 2. Results

### 2.1. Characterization of Pectin Gel Beads

Hydrogel beads named AP4 and HS4 were prepared from 4% solutions of AP and HS, respectively, using ionotropic gelling. Images of wet and dried gel beads are shown in Figure 1. 

The wet HS4 gel beads had smaller weights and dimensions than the AP4 gel beads (Table 1). The weight and surface area of wet HS4 gel beads were 1.5 and 1.35 times lower than AP4 gel beads, respectively. Gel beads AP4 and HS4 were spherical, as evidenced by the sphericity factor (SR, Table 1) and aspect ratio close to 1.0. 

The hardness of wet HS4 gel beads was 3.1 times higher than that of wet AP4 beads (Table 1). The mean force overtime curve obtained using the compression test of the wet AP4 and HS4 gel beads is shown in Figure 2. The water content, measured by drying wet beads to constant weight at 25 °C, was 95.4 ± 0.2 and 94.0 ± 0.5% (*p* < 0.05) in the AP4 and HS4 gel beads, respectively. The weight and surface area of dried HS4 gel beads were 1.1 and 1.4 times lower than dried AP4 gel beads, respectively (Table 1). The density of dried AP4 and HS4 gel beads was 1.31 ± 0.01 and 2.03 ± 0.02 mg/mm^3^ (*p* < 0.05), respectively. 

### 2.2. Swelling Studies

The swelling of the dried pectin gel beads incubated for 24 h in phosphate-buffered saline (PBS) and Hanks’ solution was investigated (Figure 3). The swelling behavior of AP4 gel beads was found to be pH-sensitive (Figure 3A). AP4 gel beads gradually swelled for 4 h to a level of 150%, then maintained this bead size for 24 h in PBS at pH 7.4. The swelling of AP4 beads at pH 5.0 continued after 4 h of incubation. The AP4 beads’ diameter was 2.73 ± 0.30 and 2.29 ± 0.19 mm (*p* < 0.05) after 4 h of incubation in PBS at pH 5.0 and pH 7.4, respectively. The swelling degree of AP4 gel beads in PBS at each time point was higher at pH 8.0 than at pH 7.4. We failed to measure the swelling of AP4 gel beads in PBS at pH 3.0 after 0.5 h of incubation due to degradation of the gel beads. An amorphous lump was observed after 4 h of incubation, followed by the complete disintegration of the AP4 gel beads after 24 h of incubation in PBS at pH 3.0. 

HS4 gel beads swelled in PBS at pH 7.4 to a greater extent than AP4 beads (Figure 3B). The swelling degree of HS4 and AP4 gel beads was 191 and 136%, respectively. HS4 beads swelled in PBS at pH 5.0 and pH 3.0 less than in PBS at pH 7.4. The HS4 beads’ diameter was 2.06 ± 0.14 (*p* < 0.05), 2.07 ± 0.13 (*p* < 0.05), and 2.39 ± 0.09 mm after 24 h of incubation in PBS at pH 5.0, pH 3.0, and pH 7.4, respectively. The swelling degree of HS4 gel beads after 0.5 h of incubation in PBS was higher at pH 8.0 than at pH 7.4. However, HS4 beads swelled equally in PBS at pH 8.0 and pH 7.4 after 4 and 24 h of incubation.

The AP4 gel beads had very low hardness after incubation in PBS. The hardness of AP4 gel beads was reduced 60 times compared with the initial hardness of wet gel beads (Table 2). The hardness of HS4 gel beads also significantly decreased upon incubation in PBS. The hardness of HS4 gel beads after 24 h of incubation in PBS at pH 7.4 was reduced 8.2 times compared with the initial hardness of wet gel beads (Table 2).

The pH value of PBS was found to change during the incubation of AP4 and HS4 gel beads. Incubation of both pectin gels in PBS with an initial pH of 5.0, 7.4, and 8.0 led to the acidification of PBS (Figure 4). The AP4 gel beads induced a faster decrease in pH than the HS4 gel beads. Both pectin gel beads decreased the pH of PBS from an initial value of 7.4 down to 6.8 and 6.9 during the 24 h of incubation of the AP4 and HS4 gel beads, respectively. The pH value decreased from 5.0 down to 3.5 and 4.6 in PBS with immersed AP4 and HS4 gel beads, respectively. In contrast, HS4 gel beads enhanced the initial pH value from 3.0 up to 3.5 during incubation (Figure 4B), whereas the AP4 gel beads failed to change pH 3.0 significantly (Figure 4A).

The swelling degree was lower at the incubation of AP4 gel beads in Hanks’ solution (pH 7.4) than in PBS at pH 7.4 (Figure 5A). The swelling degree was 1.3 times lower at the incubation of HS4 gel beads in Hanks’ solution (pH 7.4) than in PBS at pH 7.4 (Figure 5B).

The swelling of the pectin gel beads was accompanied by a decrease in the hardness of the beads (Figure 6). The hardness of both the AP4 and HS4 gel beads incubated in Hanks’ solution (pH 7.4) was significantly higher than that in PBS at pH 7.4 (Table 2). HS4 gel beads remained significantly harder than AP gel beads after 0.5, 4, and 24 h of incubation (Figure 6).

The swelling of the pectin gel beads in Hanks’ solution containing peritoneal macrophages was found to be lower than in cell-free Hanks’ solution. The effect of the composition of the incubating medium on the swelling of AP4 and HS4 gel beads is shown in Figure 7. The diameter of AP4 and HS4 gel beads incubated for 4 h in protein- and cell-free Hanks’ solutions was 11 and 12% lower than incubation in PBS, respectively. Adding 10% fetal bovine serum (FBS) to Hanks’ solution did not change the swelling of pectin gel beads. The addition of peritoneal macrophages to Hanks’ solution reduced the increase in bead diameter due to swelling by 11% and 23% for AP4 and HS4 gel beads, respectively. Further, adding lipopolysaccharide (LPS) to the incubation medium failed to influence the swelling degree.

The effect of the composition of the incubating medium on the hardness during swelling of AP4 and HS4 gel beads is shown in Table 3 and Figure 8. Similar to incubation for 24 h (see Table 2), the hardness of both AP4 and HS4 gel beads incubated for 4 h in Hanks’ solution (pH 7.3) was significantly higher than that in PBS at pH 7.4 (Table 3). Adding 10% FBS to Hanks’ solution did not change the hardness of pectin gel beads compared to Hanks’ solution. However, the hardness of AP4 and HS4 gel beads was higher by 56 and 23%, respectively, in the medium supplemented with 2 × 10^6^ cells/mL peritoneal macrophages compared to the cell-free Hanks’+FBS solution. Further, adding 10 μg/mL LPS to the incubation medium failed to change the bead hardness.

### 2.3. Protein Adsorption

The dynamic of bovine serum albumin (BSA) adsorption by the pectin gel beads incubated for 24 h in PBS was investigated. Figure 9 shows the effect of immersion of AP4 and HS4 gel beads on the concentration of BSA in PBS at different pHs. The protein adsorption by AP4 gel beads was found to be pH-sensitive (Figure 9A). Slight adsorption of BSA was observed when AP4 gel beads were incubated for 0.5 h in PBS at pH 7.4, and the equilibrium state was reached by 4 h of incubation. The BSA adsorption by AP4 gel beads increased dramatically at pH 3.0 and 5.0 by 4 h of incubation. We failed to measure the BSA adsorption by AP4 gel beads in PBS at pH 3.0 after 4 h of incubation due to degradation of the gel beads. The protein adsorption by AP4 gel beads in PBS at pH 8.0 was similar to that at pH 7.4 (Figure 9A).

The protein adsorption by HS4 gel beads was less pH-sensitive than that by AP4 gel beads (Figure 9B). HS4 gel beads decreased the concentration of BSA to the same extent in PBS at pH 5.0, 7.4, and 8.0. The adsorption of BSA by HS4 gel beads in PBS at pH 3.0 increased after 4 and 24 h of incubation. However, HS4 gel beads, when incubated in PBS at pH 3.0, adsorbed less protein than AP4 gel beads (Figure 9B).

The protein adsorption calculated per unit surface area by HS4 gel beads was 6.3 times lower, equal, and twofold higher than the AP4 gel beads at pH 5.0, 7.4, and 8.0, respectively (Figure 10).

We elucidated the adsorption of BSA by pectin gel beads in an acetate buffer to neutralize the effect of the changed pH on protein adsorption. The pH values were maintained at the initial levels of 3.7 and 5.0, respectively, when pectin gel beads were incubated in acetate buffers for 24 h (Figure 11). 

The AP4 gel beads did not degrade but gradually swelled and preserved their shape during incubation in an acetate buffer solution of pH 3.7 (data not shown). The BSA adsorption by AP4 gel beads was remarkably enhanced after 4 h of incubation in an acetate buffer solution of pH 3.7 (Figure 12A). Similar to PBS, HS4 gel beads adsorbed significantly less BSA than AP4 gel beads during incubation in acetate buffer at pH 3.7 (Figure 12A). Both pectin gels adsorbed a relatively low amount of BSA in an acetate buffer of pH 5.0 during 24 h of incubation (Figure 12B).

Pectin gel beads adsorbed a low amount of serum proteins during incubation in Hanks’ solution supplemented with 10% FBS (Figure 13).

The initial serum protein content in the Hanks’ solution supplemented with 10% FBS was 3.9 ± 0.03 mg/mL. The protein content in the supernatant obtained after incubation of AP4 and HS4 gel beads for 24 h in this medium was 3.14 ± 0.26 and 3.51 ± 0.17 mg/mL, respectively. Serum proteins adsorption by AP4 gel beads was calculated as 26 ± 19 and 196 ± 68 μg/cm^2^ after 4 and 24 h of incubation, respectively (Figure 14A). Serum proteins adsorption by HS4 gel beads was 26 ± 17 and 118 ± 44 μg/cm^2^ after 4 and 24 h of incubation (Figure 14B), indicating lower protein adsorption than AP4 gel beads.

### 2.4. Biocompatibility of Pectin Gel Beads

#### 2.4.1. Haemolysis Assay

The level of haemolysis induced by incubation of whole human blood with AP4 and HS4 gel beads is shown in Table 4. The AP4 and HS4 gel beads at a concentration of 8 mg/mL induced haemolysis ratio equal to 1.3 and 1.4%, respectively.

#### 2.4.2. Complement Activation

The release of the C3a complement component was analyzed in human blood after co-incubation with AP4 and HS4 gel beads. Both pectin gels promoted a slight, but significant, release of C3a compared to the saline samples taken as a negative control. The levels of C3a in the blood samples incubated with AP4 and HS4 gel beads were 1.4 and 1.5-fold higher than those of the blood samples with saline (Figure 15). However, the levels of C3a were significantly reduced for both pectin gels compared to zymosan-treated blood samples (positive control). The C3a levels of the blood samples incubated with HS4 gel beads were similar to the levels induced by AP4 gel beads.

#### 2.4.3. Peritoneal Macrophages Adhesion and Activation

The AP4 and HS4 gel beads were co-incubated with a suspension of mouse peritoneal macrophages in Hanks’ solution supplemented with 10% FBS and 10 μg LPS. The LPS concentration was halved after 4 h in the control cell suspension containing no pectin gel beads, probably due to the utilization of LPS by macrophages (Figure 16A). LPS content decreased 1.6 and 3.2 times compared to control in macrophage suspension co-incubated with AP4 and HS4 gel beads, respectively (Figure 16A), suggesting adsorption of LPS by the gel beads. The LPS adsorption expressed per unit surface area was 28.8 ± 7.9 and 38.4 ± 9.2 μg/cm^2^, respectively, for AP4 and HS4 gel beads incubated for 4 h in Hanks’ solution supplemented with 10% FBS and 10 μg LPS. The number of macrophages adsorbed by the beads was calculated as the difference between the initial number of cells and the number of cells remaining free in the culture medium after co-incubation for 4 h. Peritoneal macrophages were found to adhere to both AP4 and HS4 gel beads during co-incubation for 4 h in Hanks’ solution supplemented with 10% FBS, 10 μg LPS, and 2 × 10^6^ cells/mL suspension (Figure 16B). AP4 and HS4 gel beads were found to adhere to 23.5 ± 19.7 and 36.6 ± 20.6% (n = 12, *p* < 0.05) of macrophages after 4 h of incubation in the cell suspension. Macrophage adhesion expressed per unit surface area to HS4 gel beads was calculated to exceed that of the AP4 beads by 62% (Figure 16B).

The level of the proinflammatory cytokine tumor necrosis factor-α (TNF-α) in the cell supernatant was an indicator of the degree of activation of the peritoneal macrophages. AP4 gel beads enhanced the production of TNF-α by peritoneal macrophages during incubation for 4 h in Hanks’ solution supplemented with 10% FBS and 10 μg/mL LPS (Figure 17A). HS4 gel beads decreased the TNF-α level by 17 and 31% concerning control and AP4 gel beads, respectively. The enhancement of the TNF-α level induced by AP4 gel beads was accompanied by a 1.44 fold increase in the expression of Toll-like receptor 4 (TLR4) (Figure 17B,C). Moreover, macrophages expressed significantly more nuclear factor kappa B (NF-κB) in the presence of AP4 gel beads (Figure 17B,D). In contrast to AP4, HS4 gel beads decreased expression of TLR4 and NF-κB proteins by macrophages (Figure 17). 

Using the MTT assay, the percentage of cell viability of AP4 and HS4 gel beads was 87.0 ± 2.8% and 102.4 ± 7.8%, respectively.

## 3. Discussion

Hydrogel beads were prepared by simple dropwise adding 4% solutions of commercial apple pectin (AP) and pectin from hogweed *H. sosnówskyi* (HS) to calcium chloride solution. The weak mechanical properties of gels from AP in physiological environments limit their use as biomaterials. In this study, we hypothesized that HS pectin hydrogel would be more stable at swelling and exhibit better biocompatibility than AP hydrogel. Hogweed pectin was earlier found to have a lower DM than AP and, therefore, formed stronger gels in the presence of calcium ions [28]. According to the previous data, the obtained HS4 gel beads had a higher density and hardness than AP4 gel beads.

### 3.1. Swelling of Pectin Gel Beads

Swelling is a significant factor in the stability and adsorption properties of hydrogels. In this study, the swelling behavior of the pectin gel beads in PBS of different pHs and Hanks’ solution supplemented with serum proteins (FBS), macrophages, and LPS was investigated. The pectin gel beads swelled in PBS depending on pH, and the swelling degree of both AP4 and HS4 gel beads was lower in Hanks’ solution than in PBS and lower in Hanks’+FBS+macrophages than in cell-free Hanks’+FBS medium. HS4 gel beads swelled to a greater extent but were more resistant than AP4 beads in PBS and Hanks’ solutions. 

The AP4 gel beads collapsed in PBS at pH 3.0 and swelled more at pH 7.4 and 8.0 than at pH 5.0; the apparent pKa of D-GalA is 3.5 [29]. Therefore, the destruction of AP4 gel at pH 3.0 may be associated with the protonation of carboxyl groups and loss of their cross-linking by calcium ions. In addition, calcium ions that crosslink the hydrogel matrix are displaced with sodium, potassium, and phosphate ions of PBS, resulting in rapid water uptake and hydrogel erosion [30]. The leaching of calcium ions from the pectin gel was ten times higher during the swelling in the Tris-HCl buffer solution than in the Sorensen’s and McIlvaine’s buffers, containing di- and mono-hydrophosphate ions and citric acid [31]. The maximal decrease of calcium content (by 40%) was detected in our previous study in the AP gel microparticles after their incubation in an acidic medium containing sodium, potassium, and phosphate ions with pH 1.25 [32]. The crosslinking of ionized carboxyl groups of pectin chains with calcium ions at pH values higher than pKa values increases gel stability, whereas electrostatic repulsion of pectin chains promotes swelling [33]. Such swelling behavior was previously shown for pectin/acrylamide hydrogels in phosphate buffer solutions when gels swelled greater in a solution at pH 7.4 than at pH 1.2 and pH 5.4 [34,35].

HS4 gel beads swelled less in PBS at pH 3.0 and 5.0 than at pH 7.4 and 8.0. Compared to AP4, HS4 gel beads swelled equally in PBS at pH 5.0 and 8.0 and more at pH 7.4. In addition, HS4 gel beads did not break down in PBS at pH 3.0 and retained a significantly higher hardness than AP4 gel beads at all pH values. The higher hardness indicates a packed network and a higher level of interpenetration and entanglement of the pectin chains in the HS4 gel. These findings appeared to be an expected outcome as HS4 gel is stronger than AP4 gel due to the lower DM of HS than AP pectin (21 vs. 43). Increasing the number of free carboxyl groups creates more crosslinking points in the HS4 gel, reduces disintegration due to electrostatic repulsion, and increases resistance to calcium ion leaching [36,37].

The decrease in pectin gel beads swelling in Hanks’ solution appears to be associated with the presence of calcium ions (1.3 mM), which are absent in PBS. It is possible that calcium ions can diffuse from the Hanks’ solution and form additional cross-links between pectin chains, strengthening the gel. However, the content of calcium ions inside the gel beads was not measured in the present study.

The presence of cells was found to change the swelling behavior of the pectin gel. Both AP4 and HS4 gel beads swelled less in Hanks’ solution containing peritoneal macrophages than in cell-free Hanks’ solution. This effect has not been previously demonstrated, and, therefore, its mechanism remains unclear. We found that the number of cells in the macrophage suspension decreased after co-incubation with pectin gel beads, indicating cell adhesion to the beads. Therefore, we assumed that the cells adhered to the gel surface limited the intense ion exchange with the medium at swelling. It is assumed that cells cannot attach to the hydrophilic surface of a pectin hydrogel. Generally, the additional introduction of specific biochemical moieties into an otherwise bioinert pectin gel is required to enhance its functionality [38].

The data obtained indicate that the surface of the pectin gel beads may contain cell-adhesive sites. However, a limitation of our investigation should be noted in that cell adhesion to the gel was calculated but not directly observed. 

An interesting finding of the study was the shift in pH of PBS during incubation with pectin gels. The acidification of PBS with an initial pH of 5.0, 7.4, and 8.0 induced by AP4 and HS4 gel beads may be associated with shifting the equilibrium between di- and mono-hydrophosphate ions due to the formation of calcium hydrogen phosphate resulting in significant leaching of calcium ions from the gel. On the contrary, the protonation of carboxylic groups of pectin chains under acidic conditions may explain the increased initial pH of 3.0 induced by HS4 gel beads.

### 3.2. Protein Adsorption by Pectin Gel Beads

Protein adsorption is one of the first events when a biomaterial construct reacts with biological tissue [39]. Although the proteins present on most investigated surfaces include multi-protein systems including albumin, fibrinogen, immunoglobulins, vitronectin, etc., an understanding of the adsorption of a single protein is essential to understand the adsorption behaviour of individual gel materials. We used BSA to evaluate the protein adsorption due to its chemical similarity to human serum albumin. 

Similar to swelling, the adsorption of BSA by pectin gel beads was pH dependent. The electrostatic interactions might be concluded from the effects of pH on the adsorption process. BSA displays multiple positively and negatively charged surface patches. In solutions with a pH value above the pI of BSA (=4.7), BSA molecules exhibit a net negative charge, while the net charge is positive below the pI of BSA [40]. Based on these observations, the surface charge of pectin gel beads is expected to be negative above pH 3.6, according to the pKa value of free carboxyl groups of D-GalA [29]. AP4 and HS4 gel beads adsorbed a low amount of BSA at pH levels above 4.7, particularly in PBS at pH 7.4 and 8.0, as well as in acetate buffer solution at pH 5.0. Electrostatic repulsion is a reasonable explanation of these results due to both the surface of pectin gel beads and protein are expected to exhibit negative net charges. 

In PBS with an initial pH of 7.4, pectin gel beads were still able to adsorb 1.7 and 1.2 µg of BSA per cm^2^, indicating a contribution other than electrostatic interaction forces. First, the intermolecular hydrogen bonding between hydroxyl groups of pectin gels and BSA may be another driving force to adsorb BSA, as it was supposed for chitosan-based magnetic beads [41]. Moreover, the finding is likely to be explained by the high ionic strength of PBS used in the present study. Under high ionic strength conditions at pH 7.4, electrostatic interactions between BSA and AP4 and HS4 gel beads are weakened relative to van der Waals attraction, resulting in a much lower barrier to adsorption. As a result, the adsorption capacity of pectin gels remained above zero even at pH > pI, when the pectin gel surface and protein had similar charges. This observation is consistent with previous reports that describe the effect of ion strength on protein adsorption [42,43].

The maximum adsorption capacity of pectin gel beads for BSA was observed around a pH below pI of BSA when BSA is expected to be positively charged, namely, in an acetate buffer solution at pH 3.7. In addition, the adsorption capacity of HS4 gel beads increased with increasing pH from the initial level of 3.0 to 3.5 by 24 h of incubation. These findings are in good agreement with adsorption behavior that was found for alginate-based beads when the maximum adsorption capacity was reported at an acidic pH of 4.6 [44]. Pectins were previously found to remain negatively charged until reaching pH 1.5 during the titration, despite the protonation of carboxyl groups of pectin under pH below 3.6 (pKa). At pH below 4.7 (pI of BSA), low methoxy citrus pectin (DM 29) had a zeta potential of about −50 mV [45]. The electrostatic attractions can also be a reasonable explanation for the higher adsorption of BSA by HS4 gel beads at pH 5.0 than at 7.4 after 24 h because the pH drops from 5.0 to 4.6 during incubation. Electrostatic repulsion appears to regulate the low adsorption capacity of both pectin gel beads under pH-controlled conditions in acetate buffer solution at pH 5.0. 

Hanks’ solution supplemented with FBS was then used to investigate the protein adsorption capacity of pectin gel beads in relation multi-component protein system. The serum protein content in the Hanks’ solution with 10% FBS was 40 times higher than in PBS with BSA (3.9 vs. 0.2 mg/mL). Therefore, it was expected that protein adsorption by pectin gel beads in Hanks’ solution was significantly greater than in PBS. Notably, the adsorption of protein by gel beads in Hanks’ solution was 98-115 times greater than in PBS (118–196 vs. 1.2–1.7 μg/cm^2^ after 24 h), indicating that other reasons than higher protein concentrations might provide this enhancement. We suppose that the interaction between the negatively charged gel surface and serum proteins may be improved by the calcium ions from Hanks’ solution. The suggestion agrees with data that adsorption of BSA onto hydroxyapatite is governed by calcium ions complexing with protein molecules [46]. In addition, a recent study reported that the calcium ion concentration in the medium decreased during the BSA adsorption onto octacalcium phosphate materials [47]. However, the concentration of calcium ions was not analyzed during the incubation of pectin gel beads in Hanks’ solution in the present study. 

The main result of protein adsorption studies was that HS4 gel beads adsorbed serum protein less than AP4 gel beads. The difference in adsorption behavior of AP4 and HS4 pectin gels can be related to the gel properties, which are in turn determined by the structural features of the pectin macromolecule. The lower DM of HS pectin (21 vs. 43 in AP) may predispose to the higher hydrophilicity of HS4 gel beads that results in retaining bound water on the surface. Therefore, the lower protein adsorption capacity of HS4 gel beads may be due to the deep “coat” of water molecules surrounding the gel beads, which may inhibit interactions with the protein. The high DM of pectin was previously described as exhibiting better interactions with protein during the formation of complex coacervates [45]. The authors reported that the higher number of methyl ester groups shields the carboxylate anions of the pectin and thereby reduces the repulsive interaction of the pectin–protein. Differences in the structural characteristics of pectin gels and their surfaces may also be substantial, in addition to the impact of the functional groups of AP and HS pectins on protein adsorption. In light of this consideration, the higher strength of HS4 pectin gel beads during incubation in all tested media may be related to the prevention of the diffusion of protein through the HS4 gel mesh. The textural properties of pectin gel beads may indirectly reflect that HS4 gels have a denser gel network with smaller pores compared to the microstructure of AP4 gel. Therefore, the protein diffusion is expected to be lower for HS4 gel beads. This suggestion is consistent with the results reported by Yang et al. [48], which demonstrated that the samples of collagen scaffolds that had a longer diffusion distance through the hydrogel mesh exhibited higher adsorption of serum proteins. Moreover, the alginate-based biosynthetic hydrogel of a higher strength was reported to adsorb more serum proteins than the less mechanically robust hydrogel samples [49]. Similar results were obtained in our group when a pectin gel with a rough surface with globular domains adsorbed more serum proteins than the sample with a smooth surface [19]. The lower unspecific protein adsorption by the HS4 gel beads indicates a better biocompatibility than the AP4 gel beads.

### 3.3. Biocompatibility of Pectin Gel Beads

Foreign body response (FBR), which results in poor integration of implants with native tissue, represents the significant obstacle limiting biomaterials, including hydrogels. The current concept of the FBR against biomaterials divides it into phases of protein adsorption and complement activation, activation of polymorphonuclear leukocytes and macrophages, foreign body giant cell formation, and fibrosis or fibrous capsule formation [27]. In the present study, the effect of pectin gel beads on hemolysis, complement, and macrophage activity was evaluated in vitro to estimate its influence on the initial phases of FBR.

The haemocompatibility was evaluated using the measurement of haemoglobin release after incubation of pectin gel beads with whole human blood. According to the ASTM standard (*Standard Practice for Assessment of Hemolytic Properties of Materials*, ASTM F756, 2017), excessive haemolysis suggests poor erythrocyte compatibility of blood-contacting materials; therefore, the haemolysis ratio of biomaterials should be lower than 5%. The haemolysis ratio of AP4 and HS4 gel beads was lower than 1.5%, showing a negligible haemolitic effect of both pectin gels. A pectin gel prepared by mixing a heated pectin solution and a frozen calcium chloride solution induced a 4.8 ± 0.7 haemolysis rate [19]. Many other studies have previously shown good haemocompatibility for pectin-based biomaterials [50,51,52].

The haemocompatibility of biomaterials is largely determined by their ability to activate the complement system. The complement system includes a complex of serum proteins, the activation of which contributes to the recognition of foreign bodies and the induction of an innate immune response. The formation of an active C3a fragment upon contact of the serum C3 protein with a foreign surface reflects complement activation by the alternative pathway [53]. Here, the release of C3a was measured in human blood incubated with the pectin gel beads in vitro to analyze the capacity of the hydrogels to activate the alternative complement cascade. Both pectin gels promoted a slight, but significant, release of C3a compared to the saline samples used as a negative control. However, the levels of C3a were significantly reduced for both pectin gels compared to zymosan-treated blood samples used as a positive control. These results agree with the data that pectins interact directly with the C3 complement proteins [54,55,56]. 

Stimulation of inflammatory neutrophils and macrophages following nonspecific protein adsorption and activation of complement proteins reduces the functionality of the implant and may cause its disintegration [57]. Here, mouse peritoneal macrophages were treated with the pectin gel beads in the Hanks’ solution supplemented with FBS and LPS. The most important result of this study is that macrophages produced significantly less TNF-α in the presence of HS4 gel beads. A decrease in the TNF-α production by macrophages was accompanied by a decline in the expression of TLR4 and NF-κB. In contrast, AP4 gel beads enhanced the production of TNF-α and increased the expression of TLR4 and NF-κB by macrophages. Thus, the initial FBR against HS4 and AP4 was significantly different.

Pectin gel beads were found to decrease the concentration of LPS and macrophages in the medium, indicating their adsorption and adhesion, respectively. HS4 gel beads adsorbed 1.3 times more LPS and adhered to 1.6 times more cells than AP4 gel beads. LPS, or endotoxin, is an obligatory component of the outer membrane of gram-negative bacteria that activates macrophages and promotes the production of a variety of pro-inflammatory proteins, such as TNF-α [58]. LPS is one unintentional contaminant that can contribute to failed biomaterial implantation due to induction of inflammation milieu. Inflammatory response of macrophages to LPS involves transcriptional mechanisms, the most studied of which are those dependent on NF-κB, elicited by LPS binding to TLR4. LPS is composed of hydrophilic heteropolysaccharide chains and a covalently bound lipid A. Therefore, LPS represents an amphiphilic substance containing both anionic and hydrophobic groups with pK1 = 1.3 and pK2 = 8.2 [59]. In aqueous solutions, LPS aggregated into a supramolecular assembly where the lipid A moiety was proposed to be inserted into the aggregate interior, with the hydrophilic sites as head groups [59]. As the Hanks’ solution had pH 7.4, both pectin gel and LPS exhibited negative charges due to their carboxyl and phosphate groups, respectively. Therefore, electrostatic repulsion is expected to hamper interaction between LPS and pectin gel. The hydrophobic interactions and van der Waals forces, presumably, may dominate the repulsion action between LPS and the gel surface. However, the mechanism of LPS adsorption by pectin gel beads was not studied here. Although the mechanism of LPS-pectin gel interaction remains unclear, the data obtained suggests that LPS adsorption by HS4 gel beads can reduce its availability for free macrophages in the medium, thereby reducing their inflammatory activation. In addition, the inflammatory reactivity of macrophages attached to HS4 gel is likely reduced. Macrophages were earlier shown to adhere to the surface with immobilized pectin fragments [60]. The high level of Ara and Gal residues in the fragments was associated with inhibition of the LPS-induced activation of macrophages, including TNF-α secretion. Both the AP and HS pectins used in the present study consist mainly of the HG region. However, the content of Ara and Gal residues in HS pectin is 1.9 times higher than in AP. 

The inflammatory stimulation caused by AP4 gel beads may be related to an activating effect on macrophages of calcium ions, which are released from the gel when it swells. In addition, cellular debris, which results from the cytotoxic effect of AP4 gel beads, can serve as an inflammatory trigger for macrophages. According to the MTT test, 13% of the cells died during incubation with AP4 gel beads. In any case, inflammatory stimulation of macrophages is an obvious disadvantage of commercial apple pectin gel. With higher non-specific protein adsorption, these data indicate poorer biocompatibility of AP4 gel beads than HS4 gel beads. Thus, the structural features of hogweed pectin provide the formation of a gel with better mechanical properties, stability, and alleviation of the inflammatory response, which confirms the study hypothesis.

## 4. Materials and Methods

### 4.1. Polysaccharides

Apple pectin AU701 was purchased from Herbstreith and Fox (Nuremberg, Germany). Pectin HS was isolated from the leaves, petioles, and stems of hogweed *H. sosnówskyi* using extraction with 0.7% aqueous ammonium oxalate as described earlier [28]. The chemical characteristics of the pectins are shown in Table 5.

The uronic acid (UA) content was determined in pectin by reaction with 3,5-dimethylphenol in the presence of concentrated sulfuric acid, and a calibration plot was constructed for D-GalA; photocolorimetry was carried out at 400 and 450 nm [61]. The number of methyl groups was determined by the previously described method [62] from a calibration plot constructed for methanol; photocolorimetry was carried out at 412 nm. The spectrophotometric measurements were performed with an Ultrospec 3000 spectrophotometer (Pharmacia Biotech, Cambridge, UK). The content of monosaccharides was detected by gas-liquid chromatography (GLC) in a Varian 450-GC (The Netherlands) chromatograph after hydrolysis of the polysaccharides and transformation of monosaccharides into the corresponding alditol acetates. The homogeneity and molecular weight of the polysaccharide samples were determined by high performance gel-permeation chromatography (HPGPC) on a chromatographic system for the analysis included an LC-20AD pump (Shimadzu, Tokyo, Japan), a DGU-20A3 degasser (Shimadzu, Tokyo, Japan), a CTO-10AS thermostat (Shimadzu, Tokyo, Japan), a RID-10A refractometer (Shimadzu, Tokyo, Japan) as a detector, a PPS SUPREMA 3000A 10 µm (8.0 mm × 300 mm) and a PPS SUPREMA 10 µm (8.0 × 50 mm) (PSS, Amherst, MA, USA). Pullulans (1.3, 6, 12, 22, 50, 110, 200, 400, and 800 kDa) were used as standards. The detailed procedure was described earlier [28]. Spectrophotometric measurements were made with an Ultrospec 3000 spectrophotometer (Pharmacia Biotech, Cambridge, UK).

The content of endotoxin in AP and HS was 2.7 and 393.4 ng/mg, respectively, as measured using kinetic chromogenic Limulus amoebocyte lysate-test (Charles River Endosafe, INC, Reno, NV, USA). Lipopolysaccharide of *E. coli* 055:B5 (Charles River Endosafe, INC, Reno, NV, USA) was used for standard calibration curve.

### 4.2. Preparation of Gel Beads

AP and HS (40 mg/mL) were dissolved in distilled water. The pectins solutions were extruded using a pump (constant speed 0.65 mL/min) drop-wise with a 0.6 mm diameter needle into calcium chloride solution with constant stirring at 25 °C. The calcium chloride solution (340 mM) containing 25% ethanol was placed in a round-bottomed bowl, which allowed for a better distribution of the beads by volume and to avoid beads sticking together in the first seconds of the formation of the gel beads. The distance from the nozzle to the calcium chloride solution was 0.5–1.5 cm. Thus, AP4 and HS4 gel beads were formed using an excess of calcium ions. The pectin gel beads formed were allowed to stand in the calcium chloride solution for 24 h, then washed on the grid three times with 25% aqueous ethanol solution and dried at 25 °C to constant weight, as described previously [63].

### 4.3. Characterization of Gel Beads

Images of dry gel beads (n = 40) were obtained using an optical microscope (Altami, Russia) equipped with a camera. The projected equivalent diameter of the beads was determined using an image analysis system (ImageJ 1.46r program, National Institutes of Health, Bethesda, MD, USA) with calibration of 0.024 mm to one pixel. 

A compression test of the gel beads was performed using TA-XT Plus Texture Analyzer (Texture Technologies Corp., Stable Micro Systems, Godalming, UK). The wet gel beads were compressed at 25 °C with a 12 mm diameter (P/0.5R) cylinder probe with the pre- and post-test speed was 10.0 mm/s and the test speed of 0.5 mm/s until deformation of 50%. The detailed procedure was described earlier [64]. The calculations of maximum peaks were performed for ten replicate samples using Texture Exponent 6.1.4.0 software (Stable Micro Systems, Godalming, UK).

The water content was calculated using the following Equation (1): (W_W_ − W_D_)/W_W_ × 100%,(1)
where W_W_ and W_D_ represent the weight of the gel beads (n = 20–30) before and after drying at 25 °C until constant weight.

### 4.4. Swelling Characterization of Gel Beads

Twenty mg of dry AP4 and HS4 gel beads (ca. 30 beads) were incubated in PBS (NaCl 137 mM, KCl 2.7 mM, Na_2_HPO_4_ 10 mM, KH_2_PO_4_ 1.8 mM), acetate buffer solutions (CH_3_COONa 150 mM, CH_3_COOH 150 mM) or Hanks’ solutions for 2, 4, and 24 h with shaking (100 rpm) in an orbital shaker incubator (Titramax 1000, Heidolph, Germany) at 37 °C. Hanks’ solution was used in the following four variations: Hanks’ balanced salts (NaCl 140 mM, KCl 5 mM, CaCl_2_ 1 mM, MgSO_4_ 0.4 mM, MgCl_2_ 0.5 mM, Na_2_HPO_4_ 0.3 mM, KH_2_PO_4_ 0.4 mM, D-glucose 6 mM, NaHCO_3_ 4 mM, pH 7.4) solution; Hanks’ solution supplemented with 10% FBS; Hanks’ solution supplemented with 10% FBS and 2 × 10^6^ cells/mL peritoneal macrophages; Hanks’ solution supplemented with 10% FBS, 2 × 10^6^ peritoneal macrophages, and 10 μg/mL of LPS. Each Hanks’ solution variant was supplemented with 25 mM HEPES to support constant pH 7.4. After a predetermined time interval, the projected equivalent diameter of beads was measured using an optical microscope (Altami, Russia) fitted with a camera and an image analysis system (Altami Studio, Altami, Russia). An image of a linear scale was used for calibration under the same optical conditions. One pixel corresponds to 0.00593 mm. The swelling degree (SD) was calculated using the following Equation (2):SD% = ((S_1_ − S_0_)/S_0_) × 100,(2)
where S_1_ is the projected equivalent diameter of the bead after a determined incubation time and S_0_ is the initial diameter.

The hardness of pectin gel beads was determined after 0.5, 4, and 24 h of incubation using a compression test as described above.

### 4.5. Protein Adsorption by Gel Beads

The adsorption of BSA (MP Biomedicals, Eschwege, Germany) by pectin gel beads was investigated as reported previously [65,66]. Dried AP4 and HS4 gel beads (20 mg) were immersed in 10 mL of PBS of pH 7.4 containing BSA at a concentration of 0.2 mg/mL. Then pH of PBS containing BSA was immediately adjusted to pH 3.0, 5.0, and 8.0 by dropping 1N hydrochloric acid or sodium hydroxide solution before each adsorption experiment. In the pH-controlled adsorption tests, the 0.15 M acetate buffer solutions of pH 3.7 and 5.0 containing BSA at a concentration of 0.2 mg/mL were used in pH-controlled adsorption tests. Adsorption of serum proteins was investigated by immersing pectin gel beads in Hanks’ solution supplemented with 10% FBS. In all adsorption experiments, pectin gel beads were put in glass vials and incubated at 37 °C with 100 rpm shaking. Protein solutions without gel beads and the formulations of gel beads free of proteins were served as blanks for each experiment. The amount of protein in the sample aliquots was analyzed after pre-defined time intervals of 0.5, 4, and 24 h of incubation. Upon incubation, samples were centrifuged 1000× *g* for 20 min at 4 °C (Micro 220 R Hettich Zentrifugen, Tuttlingen, Germany), and the supernatant containing free protein was collected. The free protein of the supernatants was measured by the Micro BCA Protein Assay Kit (Thermo Scientific™, Waltham, MA, USA). The calibration curve was prepared by measuring varying BSA concentrations in PBS or Hanks’ solution as the absorbance at 562 nm was measured using a Power wave 200 reader (BioTek Instruments, Santa Clara, CA, USA). Based on this curve, the amount of the adsorbed protein was expressed as μg/cm^3^ and calculated using the following Equation (3):(C_0_ − C_t_)V/S,(3)
where C_0_ and C_t_ are the initial concentration of protein and the concentration of protein (mg/L) remaining in solution at different time points after incubation, respectively; V (L) is the total volume of the solution, and S (cm^2^) is the total surface area of pectin gels after removal from solution.

The adsorption kinetics was estimated as the relative protein concentration [C_0_/C_t_] when pectin gels were immersed in protein solutions during incubation.

### 4.6. Haemolysis Ratio Determination

The hemolysis ratio was measured after incubation of whole blood with pectin gel beads. The dried gel beads at different concentrations (2, 4, and 8 mg/mL) were put into sterile 2 mL microcentrifuge tubes (Eppendorf, Leipzig, Germany), and 0.3 mL of blood was added to each tube and incubated for 1 h at 37 °C without stirring. Following incubation, 0.1 mL of the supernatants of whole blood were collected after centrifugation at 400× *g* for 20 min at 4 °C (Micro 220R Hettich Zentrifugen, Tuttlingen, Germany), and the OD was measured at 540 nm using a Power wave 200 reader (BioTek Instruments, Santa Clara, CA, USA). The distilled water (0.1 mL) and pyrogen-free 0.9% NaCl (0.05 mL) were used as positive and negative controls, respectively. The hemolysis ratio was calculated using the following Equation (4):Hemolysis ratio (%) = ((A_s_ − A_n_)/(A_p_ − A_n_)) × 100,(4)
where A_s_ is the absorbance of the hydrogel sample, A_p_ and A_n_ are the absorbance of the positive control and the negative control, respectively.

### 4.7. Complement Activation Evaluation

Venous blood was collected from healthy volunteers into vacuum tubes (Improvacuter, Guangzhou Improve Medical Instruments, Guangzhou, China) after obtaining written informed consent. The protocol was approved by the Ethical Committee of the Komi Science Centre of the Russian Academy of Sciences. Complement activation evaluation was performed by measuring the C3a levels as previously described [67]. To measure the release of C3a as an indicator of complement activation, 0.25 mL of human blood was added to the dry pectin gel beads at a concentration of 2 mg/mL. Zymosan A (Sigma-Aldrich, St. Louis, MO, USA) at a final concentration of 0.100 mg/mL was used as positive control and pyrogen-free 0.9% saline (NaCl, 0.05 mL) as a negative control. The blood samples were incubated at 37 °C for 2 h. Following incubation, the supernatants of whole blood were collected after centrifugation 400× *g* for 20 min at 4 °C (Micro 220R Hettich Zentrifugen, Tuttlingen, Germany). Obtained supernatants were frozen and stored at −40 °C for later analysis. An enzyme-linked immunosorbent assay (ELISA) kit (Human C3a ELISA kit, Hycult Biotech, Uden, The Netherlands) was used according to the manufacturer’s instructions to determine the concentration of released C3a fragments in plasma, which was diluted 1:1000 with C3a-Sample Diluent.

### 4.8. Peritoneal Macrophages Adhesion and Activation

Macrophages were obtained using lavage of the abdominal cavity of male BALB/c mice (25–30 g) with PBS (5 mL). The cells were centrifuged in saline for 10 min at 400× *g* and then resuspended in Hanks’ balanced solution containing 25 mM HEPES (pH 7.4) and 10% FBS. For cell response, 1 mL of 10^6^ peritoneal macrophages were incubated with 2 mg of pectin gel beads at 37 °C for 4 h in microcentrifuge tubes (Eppendorf, Leipzig, Germany). The number of adherent cells was defined as the following Equation (5): ((C0 − C4)/C0) × 100,(5)
where C0 and C4 represent the initial cell concentration and after 4 h of co-incubation with the beads, respectively. After 4 h of incubation, the cells were precipitated by centrifugation (10 min, 400× *g*), and the supernatant was separated for subsequent cytokine evaluation using ELISA as described [67]. The LPS content in supernatants was measured using kinetic chromogenic Limulus amoebocyte lysate-test (Charles River Endosafe, INC, Reno, Nevada, USA) and *E. coli* 055:B5 LPS (Charles River Endosafe, INC, Reno, NV, USA) as calibration standard. The cells were homogenized for 10 min at 4 °C in 0.2 mL of lysis buffer (50 mM Tris-HCl, 150 mM NaCl, 50 mM NaF, 5 mM EDTA, 0.1% SDS, 0.1% Triton X-100, 1 mM PMSF, 1 mM Na_3_VO_4_ and protease inhibitor cocktail (Sigma-Aldrich, St. Louis, MO, USA) at a 1:100 dilution). The samples were centrifuged at 4 °C at 10,000× *g* for 20 min and then stored at −40 °C until measurements. Western blotting analysis of NF-κB and TLR4 was performed as described earlier [67].

Cytotoxicity of gel beads was assessed by indirect MTT assay according to ISO 10993-5 using mouse peritoneal cells. The extraction medium was prepared through immersing gel beads (15 mg) in 5 mL of Hanks’ balanced salt solution supplemented with 25 mM HEPES and 10% fetal calf serum for 4 h at 37 °C. After removing gel beads, medium was used for incubation mouse peritoneal cells. Mouse peritoneal cells at density of 10^6^ per mL were incubated in 0.5 mL of the extraction medium with added MTT reagent (0.5 mg/mL) and LPS (10 μg/mL) for 1 h at 37 °C. After incubation cells were centrifuged at 4 °C at 400× *g* for 10 min. Supernantant were replaced by DMSO (0.3 mL) in order to dissolve formazan crystals. The absorbance was evaluated at 570 and 620 nm. The absorbance at 620 nm was substracted from the absorbance at 570 nm. The cell viability percentage was then calculated against the optical density of the control, mouse peritoneal cells incubated in fresh medium with added MTT reagent (0.5 mg/mL) and LPS (10 μg/mL).

### 4.9. Statistical Analysis

The results were presented as the arithmetic mean ± standard deviation. One-way ANOVA with Tukey’s honest significance test was applied to determine statistically significant differences in gel characterization, swelling, protein adsorption, haemolysis, and complement activation. ANOVA for repeated measurements was used to determine statistically significant differences in experiments that involved mouse peritoneal cells. Values of *p* ≤ 0.05 were considered statistically significant.

## 5. Conclusions

In this study, the advantages of HS pectin gel compared to AP gel are demonstrated concerning swelling behavior, protein adsorption, and biocompatibility. HS4 and AP4 gel beads swelled in PBS with a dependence on pH and swelled less in Hanks’ solution than in PBS. Both pectin gel beads swelled less in Hanks’ solution containing peritoneal macrophages than in cell-free Hanks’ solution. HS4 gel beads swelled to a greater extent but were more resistant than AP4 beads in all tested media. HS4 gel beads adsorbed serum protein less than AP4 gel beads. Both pectin gel beads demonstrated low rates of hemolysis and complement activation. However, HS4 gel beads inhibited the LPS-stimulated secretion of TNF-α and the expression of TLR4 and NF-κB by macrophages, whereas AP4 gel beads stimulated the inflammatory response of macrophages. The anti-inflammatory effect of HS4 gel beads may be related to the adsorption of LPS and the adhesion of macrophages. Thus, pectin hydrogel was found to change its properties under exposure to macrophages in addition to pH- and ion composition- sensitivity. Hogweed pectin has unique structural and gel-forming properties, making it a promising candidate for smart gel development.

## Figures and Tables

**Figure 1 ijms-23-03388-f001:**
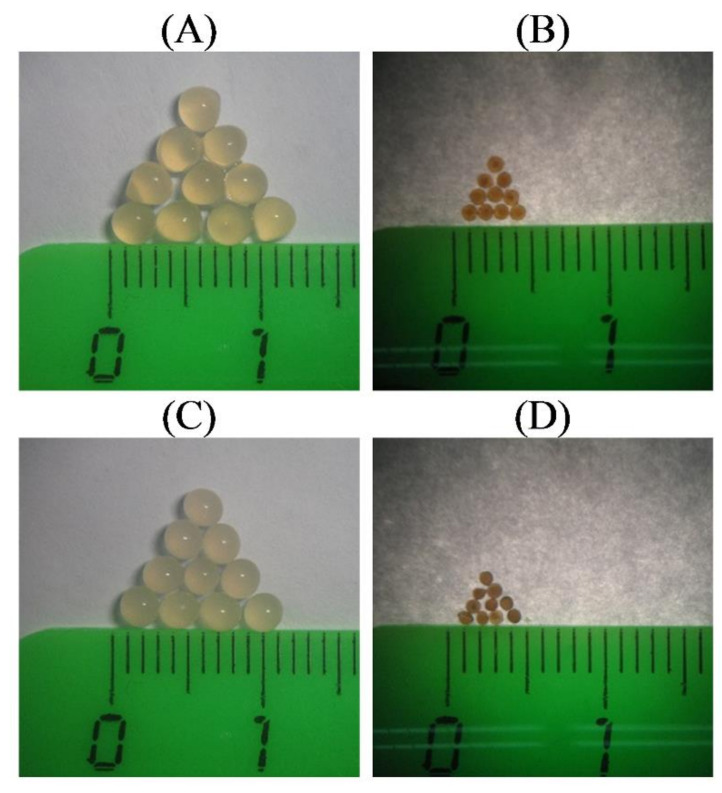
Photographs of wet AP4 (**A**), HS4 (**C,**) and dried AP4 (**B**), HS4 (**D**) gel beads.

**Figure 2 ijms-23-03388-f002:**
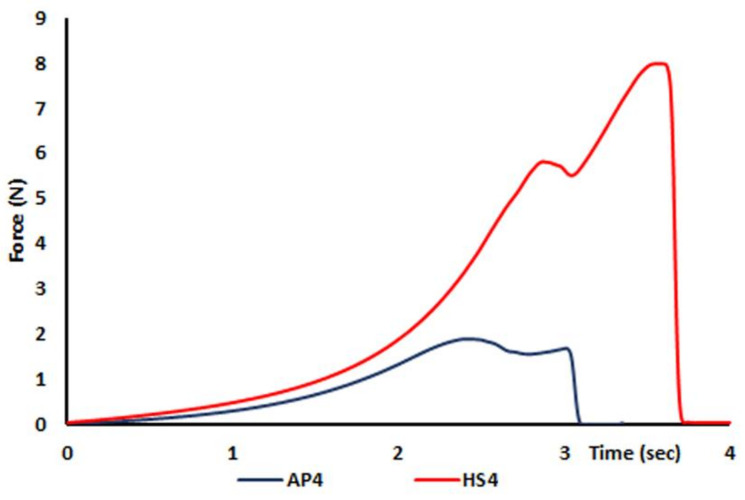
Mean (n = 15) force overtime curve of the wet AP4 and HS4 gel beads.

**Figure 3 ijms-23-03388-f003:**
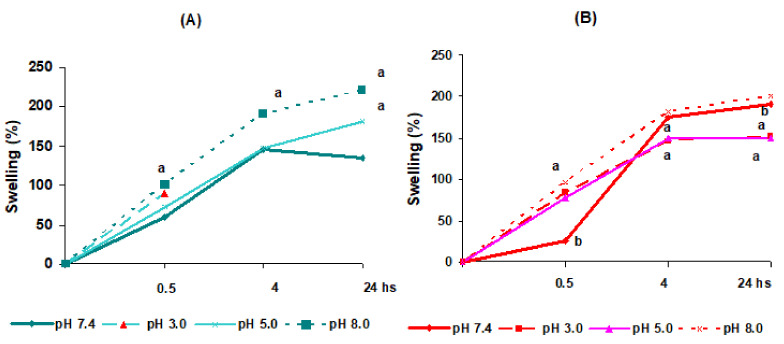
Swelling degree of AP4 (**A**) and HS4 (**B**) gel beads incubating in PBS of different pH for 24 h. The data are presented as the mean (n = 10). ^a^ *p* < 0.05 vs. PBS pH 7.4; ^b^ *p* < 0.05 vs. the corresponding AP4 gel beads.

**Figure 4 ijms-23-03388-f004:**
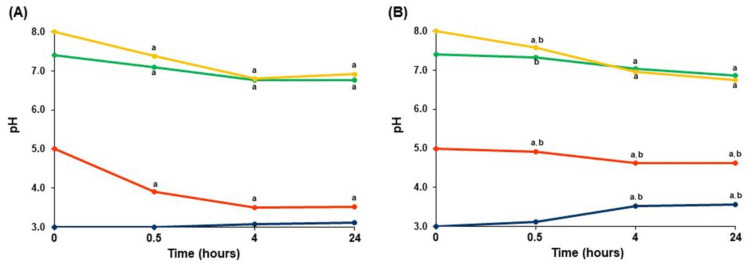
Variation in the pH values during incubation of AP4 (**A**) and HS4 (**B**) gel beads in PBS at different pH for 24 h. The data are presented as the mean ± S.D. (*n* = 5). ^a^
*p* < 0.05 vs. corresponding initial pH; ^b^ *p* < 0.05 vs. the corresponding AP4 gel beads levels.

**Figure 5 ijms-23-03388-f005:**
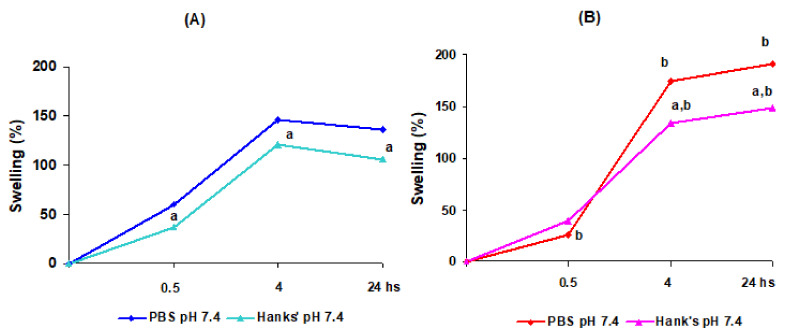
Swelling degree of AP4 (**A**) and HS4 (**B**) gel beads incubating in PBS at pH 7.4 and Hanks’ solution (pH 7.4) for 24 h. The data are presented as the mean (n = 10). ^a^ *p* < 0.05 vs. PBS pH 7.4; ^b^ *p* < 0.05 vs. the corresponding AP4 gel beads.

**Figure 6 ijms-23-03388-f006:**
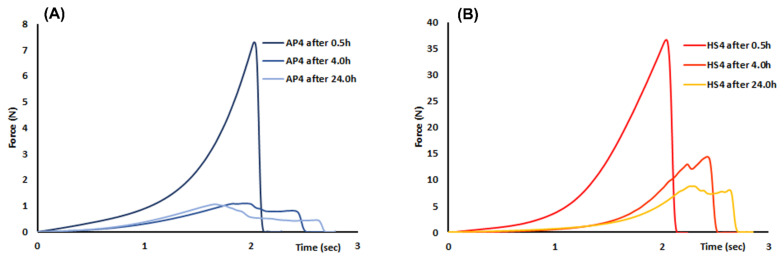
Mean (*n* = 15) force overtime curve of the AP4 (**A**) and HS4 (**B**) gel beads incubated in Hanks’ solution for 24 h.

**Figure 7 ijms-23-03388-f007:**
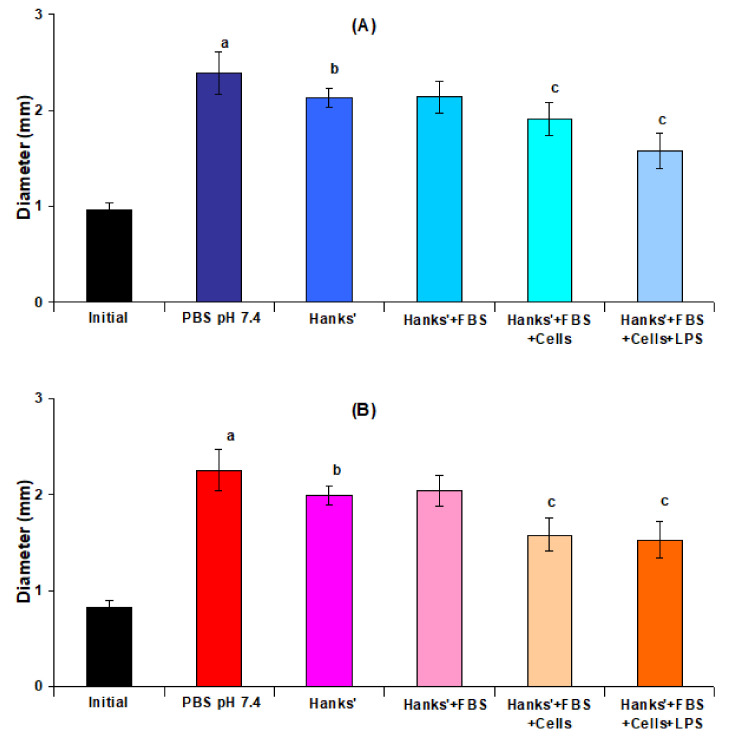
Effect of composition of incubating medium on the swelling of AP4 (**A**) and HS4 (**B**) gel beads. Dried gel beads (“Initial”) were incubated for 4 h in PBS (“PBS” pH 7.4); Hanks’ solution at pH 7.4 (“Hanks’”); Hanks’ solution supplemented with 10% fetal bovine serum (“Hanks’+FBS”); Hanks’ solution supplemented with 10% fetal bovine serum and 2 × 10^6^ cells/mL peritoneal macrophages (“Hanks’+FBS+Cells”); Hanks’ solution supplemented with 10% fetal bovine serum, 2 × 10^6^ peritoneal macrophages, and 10 μg/mL of lipopolysaccharide (“Hanks’+FBS+Cells+LPS”). The data are presented as the mean ± S.D. (n = 12). ^a^ *p* < 0.05 vs. “Initial”; ^b^ *p* < 0.05 vs. PBS pH 7.4; ^c^ *p* < 0.05 vs. Hanks’+FBS.

**Figure 8 ijms-23-03388-f008:**
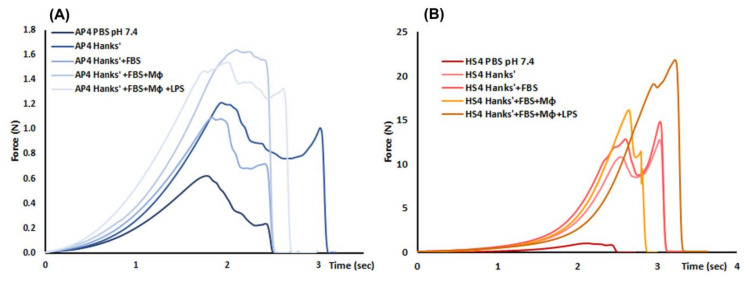
Mean (*n* = 15) force overtime curve of the AP4 (**A**) and HS4 (**B**) gel beads incubated in PBS (“PBS pH 7.4”); Hanks’ solution at pH 7.4 (“Hanks’”); Hanks’ solution supplemented with 10% FBS (“Hanks’+FBS”); Hanks’ solution supplemented with 10% FBS and 2 × 10^6^ cells/mL peritoneal macrophages (“Hanks’+FBS+Mφ”); Hanks’ solution supplemented with 10% fetal bovine serum, 2 × 10^6^ peritoneal macrophages, and 10 μg/mL of LPS (“Hanks’+FBS+Mφ+LPS”).

**Figure 9 ijms-23-03388-f009:**
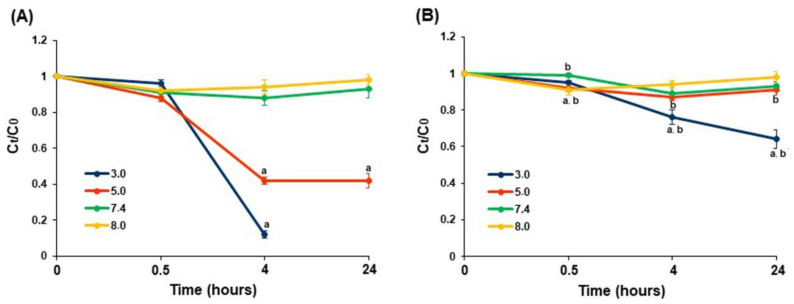
The BSA adsorption by AP4 (**A**) and HS4 (**B**) gel beads incubating in PBS at different pH for 24 h. The data are presented as the mean ± S.D. (*n* = 5). ^a^ *p* < 0.05 vs. PBS at pH 7.4; ^b^ *p* < 0.05 vs. the corresponding AP4 gel beads. The BSA adsorption is expressed as the relative protein concentration C_t_/C_0_, where C_0_ and C_t_ are the initial and remaining concentrations of BSA in PBS at different time points after incubation, respectively.

**Figure 10 ijms-23-03388-f010:**
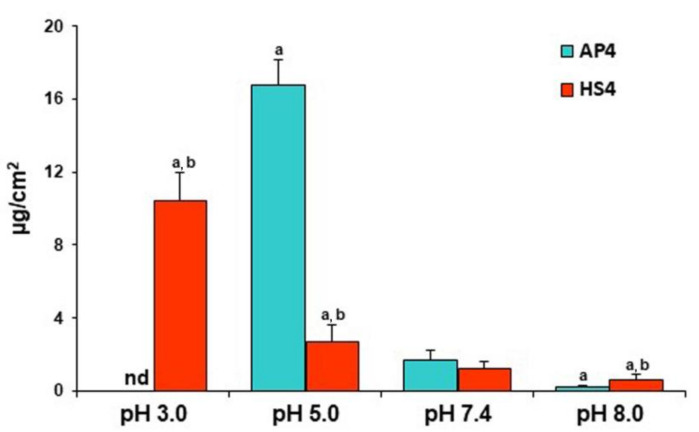
The BSA adsorption expressed per unit surface area by pectin gel beads incubating in PBS at different pH for 24 h. The data are presented as the mean ± S.D. (*n* = 5). ^a^ *p* < 0.05 vs. PBS pH 7.4; ^b^ *p* < 0.05 vs. the corresponding AP4 gel beads.

**Figure 11 ijms-23-03388-f011:**
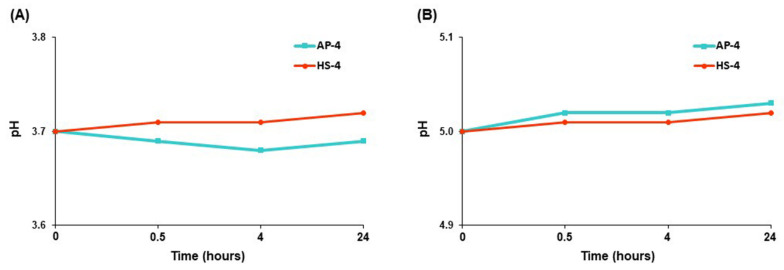
Variation in the pH values during incubation of AP4 and HS4 gel beads for 24 h in acetate buffer solutions of pH 3.7 (**A**) and 5.0 (**B**). The data are presented as the mean ± S.D. (*n* = 5).

**Figure 12 ijms-23-03388-f012:**
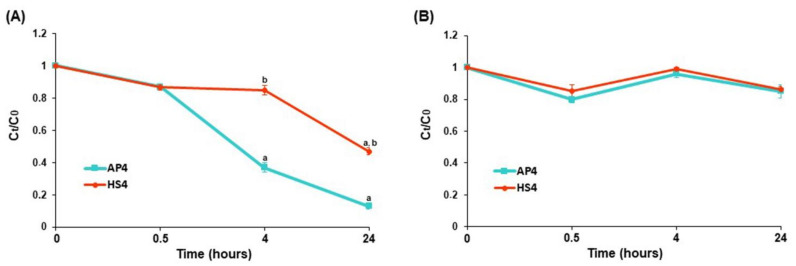
The BSA adsorption by AP4 and HS4 gel beads incubating for 24 h in acetate buffer solutions of pH 3.7 (**A**) and 5.0 (**B**). The data are presented as the mean ± S.D. (*n* = 5). ^a^ *p* < 0.05 vs. initial time point; ^b^ *p* < 0.05 vs. the corresponding AP4 gel beads. The BSA adsorption is expressed as the relative protein concentration C_t_/C_0_, where C_0_ and C_t_ are the initial and remaining concentrations of BSA in PBS at different time points after incubation, respectively.

**Figure 13 ijms-23-03388-f013:**
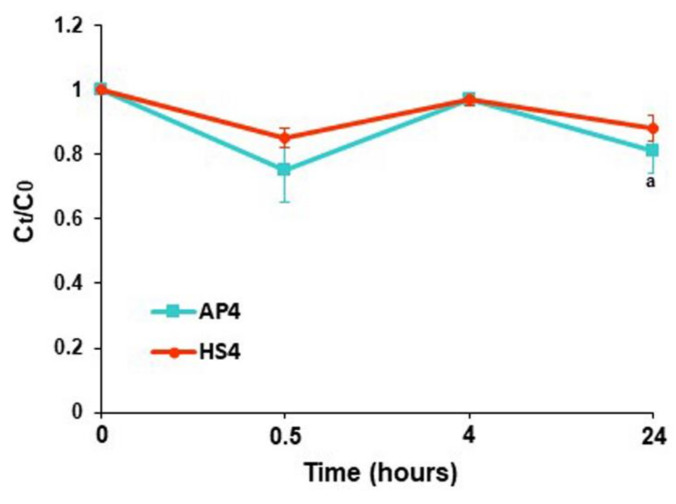
The serum proteins adsorption by AP4 and HS4 gel beads incubating for 24 h in Hanks’ solution supplemented with 10% FBS. The data are presented as the mean ± S.D. ^a^ *p* < 0.05 vs. the corresponding AP4 gel beads. The protein adsorption is expressed as the relative protein concentration C_t_/C_0_, where C_0_ and C_t_ are the initial and remaining concentrations in Hanks’ solution at different time points after incubation, respectively.

**Figure 14 ijms-23-03388-f014:**
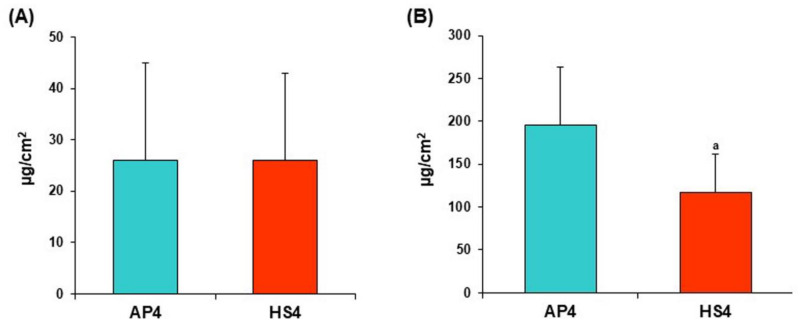
The serum proteins adsorption expressed per unit surface area by AP4 and HS4 gel beads incubating for 4 (**A**) and 24 h (**B**) in Hanks’ solution supplemented with 10% FBS. The data are presented as the mean ± S.D. (n = 5). ^a^ *p* < 0.05 vs. the corresponding AP4 gel beads.

**Figure 15 ijms-23-03388-f015:**
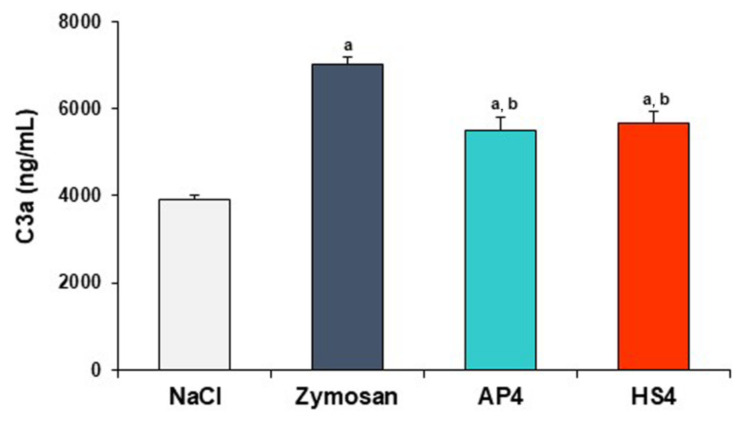
Effect of AP4 and HS4 gel beads on the C3a production in the whole blood in vitro. Results are presented as the mean ± S.D. (n = 8). ^a^ and ^b^—*p* < 0.05 vs. NaCl and Zymosan, respectively.

**Figure 16 ijms-23-03388-f016:**
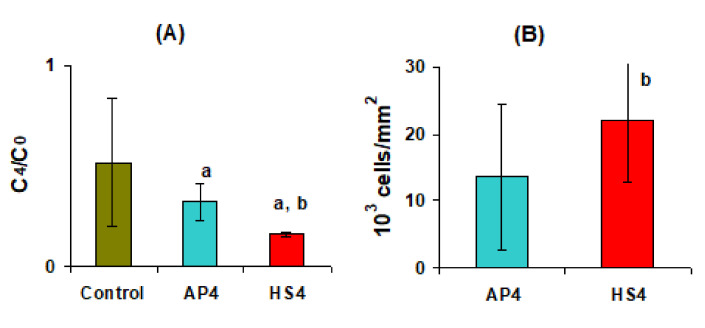
LPS adsorption (**A**) and macrophage adhesion (**B**) on the AP4 and HS4 gel beads during incubation for 4 h in Hanks’ solution supplemented with 10% FBS, 10 μg/mL LPS, and 2 × 10^6^ peritoneal macrophages. The data are presented as the mean ± S.D. (n = 12). ^a^ and ^b^—*p* < 0.05 vs. Control and AP4 gel beads, respectively. C_4_/C_0_—relative concentration, where C_0_ and C_4_ are the initial and remaining concentrations of LPS in Hanks’ solution after 4 h of incubation, respectively.

**Figure 17 ijms-23-03388-f017:**
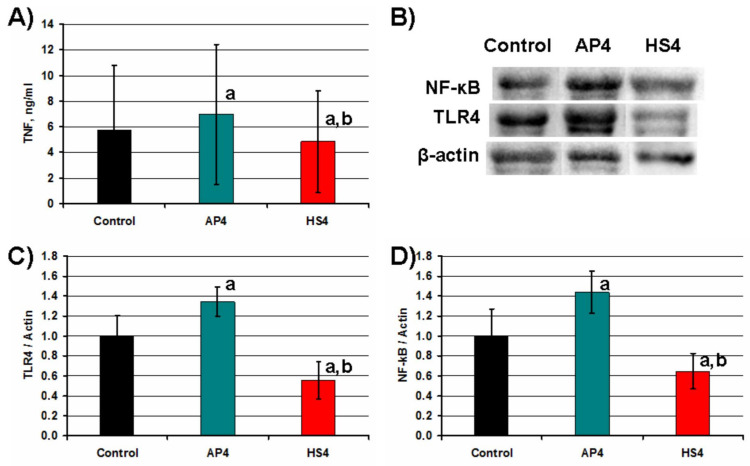
The impact of pectin gel beads on inflammatory proteins levels in mouse peritoneal macrophages. Bar graphs show TNF-α concentration (**A**) and TLR4 (**C**), and NF-κB (**D**) protein levels, which are normalized to the β-actin protein level. (**B**)—representative bands of NF-κB, TLR4, and β-actin. The results are expressed as the mean ± S.D. (n = 8). ^a^ and ^b^—*p* < 0.05 vs. Control and AP4 gel beads, respectively.

**Table 1 ijms-23-03388-t001:** Characterization of gel beads.

Gel Bead	Diameter (mm)	Weight (mg)	S * (mm^2^)	SF **	Hardness (N)
Wet AP4	2.8 ± 0.2	13.5 ± 0.2	24.6 ± 2.5	0.03 ± 0.02	1.8 ± 0.2
Wet HS4	2.4 ± 0.1 ^a^	9.0 ± 0.2 ^a^	18.2 ± 1.6 ^a^	0.01 ± 0.01 ^a^	5.7 ± 0.9 ^a^
Dried AP4	0.97 ± 0.07	0.64 ± 0.002	2.97 ± 0.04	0.04 ± 0.03	n.d.
Dried HS4	0.82 ± 0.08 ^a^	0.60 ± 0.006 ^a^	2.13 ± 0.04 ^a^	0.04 ± 0.03	n.d.

*—Surface area; **—Sphericity factor; n.d.—not determined. The data are presented as the mean ± standard deviation (S.D.) (*n* = 15). a *p* < 0.05 vs. the corresponding AP4 gel beads.

**Table 2 ijms-23-03388-t002:** The hardness (N) of pectin gel beads after 24 h incubation in PBS and Hanks’ solution.

Gel Bead	PBS pH 3.0	PBS pH 5.0	PBS pH 7.4	PBS pH 8.0	Hanks’ (pH 7.4)
AP4	n.d.	0.04 ± 0.01^b^	0.34 ± 0.10	0.26 ± 0.13 ^b^	1.01 ± 0.12 ^b^
HS4	3.30 ± 0.34 ^b^	4.85 ± 0.53 ^a,b^	0.69 ± 0.09 ^a^	1.64 ± 0.20 ^a,b^	8.82 ± 0.74 ^a,b^

The data are presented as the mean ± S.D. (n = 15). ^a^ *p* < 0.05 vs. the corresponding AP4 gel beads; ^b^ *p* < 0.05 vs. PBS pH 7.4. n.d.—not determined.

**Table 3 ijms-23-03388-t003:** The hardness (N) of pectin gel beads after 4 h incubation in different media.

Gel Bead	PBS pH 7.4	Hanks’	Hanks’+FBS	Hanks’+FBS+Cells	Hanks’+FBS+Cells+LPS
AP4	0.60 ± 0.11	1.19 ± 0.25^b^	1.01 ± 0.12	1.58 ± 0.34 ^c^	1.55 ± 0.34
HS4	0.94 ± 0.16 ^a^	10.91 ± 1.24 ^a,b^	12.30 ± 2.33 ^a^	15.10 ± 5.28 ^a,c^	18.20 ± 3.20 ^a^

Abbreviations are given as in Figure 6. The data are presented as the mean ± S.D. (n = 15). ^a^ *p* < 0.05 vs. the corresponding AP4 gel beads; ^b^ *p* < 0.05 vs. PBS pH 7.4; ^c^ *p* < 0.05 vs. Hanks’+FBS.

**Table 4 ijms-23-03388-t004:** Effect of pectin gel beads on haemolysis of whole blood in vitro.

Samples, Concentrations	OD (540 nm)	Haemolysis Ratio (%)
Distilled Water (Positive control)	3.992 ± 0.107 ^a^	100 ± 0
0.9% NaCl (Negative control)	0.099 ± 0.003	0 ± 0
AP4 2 mg/mL	0.110 ± 0.008	0.3 ± 0.2
AP4 4 mg/mL	0.116 ± 0.004 ^a^	0.5 ± 0.1 ^a^
AP4 8 mg/mL	0.156 ± 0.016 ^a^	1.3 ± 0.4 ^a^
HS4 2 mg/mL	0.115 ± 0.009	0.4 ± 0.2
HS4 4 mg/mL	0.124 ± 0.007 ^a^	0.7 ± 0.2 ^a^
HS4 8 mg/mL	0.155 ± 0.014 ^a^	1.4 ± 0.03 ^a^

OD—optical density. The data are presented as the mean ± S.D. (*n* = 8). ^a^ *p* < 0.05 vs. 0.9% NaCl values.

**Table 5 ijms-23-03388-t005:** Chemical characteristics of pectins.

Sample	UA ^a^	Gal ^a^	Xyl ^a^	Glc ^a^	Rha ^a^	Ara ^a^	OMe ^a^	DM	M_w_, kDa	M_w_/M_n_
HS	82.3 ± 0.6	2.7 ± 0.2	0.7 ± 0.2	0.4 ± 0.2	1.6 ± 0.1	2.3 ± 0.1	3.0 ± 0.6	21	538	4.1
AP	86.5 ± 0.7	2.3 ± 0.1	2.8 ± 0.1	1.5 ± 0.1	1.3 ± 0.1	0.6 ± 0.4	6.2 ± 0.4	43	401	5.2

^a^ Data were calculated as weight %. UA—uronic acids, Gal—galactose, Xyl—xylose, Glc—glucose, Rha—rhamnose, Ara—arabinose, OMe—the amount of methyl groups, DM—degree of methyl esterification.

## Data Availability

The data that support the findings of this study are available from the corresponding author upon reasonable request.

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
