# Peer review of "Swelling, Protein Adsorption, and Biocompatibility In Vitro of Gel Beads Prepared from Pectin of Hogweed Heracleum sosnówskyi Manden in Comparison with Gel Beads from Apple Pectin"

_ijms, 2022, doi:10.3390/ijms23063388_

Round 1

Reviewer 1 Report

Dear Authors

In your paper you forgot about the description of your methodology

Please look i.e. here – as it should be done (it is not my paper – iI found just another exemplary paper in IJMS journal)

Chitosan-Based Coacervate Polymers for Propolis Encapsulation: Release and Cytotoxicity Studies

https://doi.org/10.3390/ijms21124561

In your draft:

Material – are unknown for the reader – we know i.e. that you study apple pectin – but we don’t know the origin or pectic, purity, and other compounds parameters. The same for other compounds used in your research.

We know nothing about your methodology.

No only this – the information about your gel formation methodology – it also limited to “Hydrogel beads named AP4 and HS4 were prepared from 4% solutions of AP and 80 HS, respectively, using ionotropic gelling.” – I only know that you use 4% solution and you gel it – how long? In which temperature? Was it stirred or not? May you use some special procedure of gelling? Nobody know – and nobody can repeat your experiments.

Nobody also know your research methodology

Without precisely described methodology of protein adsorption experiments, swelling studies, biocompatibility test …. etc.etc. nobody can analyze the results of your experiments.

I have read yourdraft but I will not evaluate its value until the basic descriptions of materials and research methods necessary for each scientific text are introduced to it

Reviewer 2 Report

The manuscript entitled “Swelling, protein adsorption, and biocompatibility in vitro of gel beads prepared from pectin of hogweed Heracleum sosnówskyi Manden in comparison with gel beads from apple pectin” presents the characterization of two gel beads prepared from pectin obtained of hogweed Heracleum sosnówskyi Manden or apple followed by in vitro biocompatibility.

Although the data are well presented and described, there are some issues that need to be addressed.

First, the authors don’t have a section regarding materials and methods, which makes it harder to follow the results. The materials and methods section needs to be added.

For haemolysis it should be explained if the percentages reported are indicating a haemolytic effect or not.

The MTT assay is mentioned, but no results are showed. These should be also added in the manuscript.

In some figures the authors say there is statistical difference vs control or AP4, but considering the mean and the error bar, there should be no difference.

Round 2

Reviewer 1 Report

I accept this version of the manuscript

Reviewer 2 Report

The authors have made the changes asked and the quality of the manuscript improved. The paper can be accepted in the present format.